# Correlation between Macular Pigment Optical Density and Neural Thickness and Volume of the Retina

**DOI:** 10.3390/nu12040888

**Published:** 2020-03-25

**Authors:** Norihiro Nagai, Teru Asato, Sakiko Minami, Misa Suzuki, Hajime Shinoda, Toshihide Kurihara, Hideki Sonobe, Kazuhiro Watanabe, Atsuro Uchida, Norimitsu Ban, Kazuo Tsubota, Yoko Ozawa

**Affiliations:** 1Laboratory of Retinal Cell Biology, Keio University School of Medicine, 35 Shinanomachi, Shinjuku-ku, Tokyo 160-8582, Japan; nagai@a5.keio.jp (N.N.); misayutakatomo@icloud.com (M.S.); 2Department of Ophthalmology, Keio University School of Medicine, 35 Shinanomachi, Shinjuku-ku, Tokyo 160-8582, Japan; doratarou23@gmail.com (T.A.); saki.love5@icloud.com (S.M.); shinoha@mac.com (H.S.); kurihara@z8.keio.jp (T.K.); betty_vol2@ybb.ne.jp (H.S.); gaku047nikoniko3mickey@yahoo.co.jp (K.W.); uchidats@gmail.com (A.U.); nban@keio.jp (N.B.); tsubota@z3.keio.jp (K.T.); 3Department of Ophthalmology, St. Luke’s International Hospital, 9-1 Akashi-cho, Chuo-ku, Tokyo 104-8560, Japan

**Keywords:** macular pigment, optical coherence tomography, macula, retina, neural tissue

## Abstract

Macular pigment (MP), which is composed of lutein/zeaxanthin/mezo-zeaxanthin, is concentrated in the central part of the retina, the macula. It protects the macula by absorbing short-wavelength light and suppressing oxidative stress. To evaluate whether MP levels are related to retinal neural protection and resulting health, we analyzed the association between the MP optical density (MPOD), and the macular thickness and volumes. Forty-three eyes of 43 healthy adult volunteers (21 men and 22 women; age: 22–48 (average 31.4 ± 1.1) years) were analyzed. Highly myopic eyes (<-6 diopters) were excluded. MPOD was measured using MPS2®, and the neural retinal thickness and volume were measured using optical coherence tomography. The mean MPOD was 0.589 ± 0.024, and it positively correlated with the central retinal thickness (*P* = 0.017, R = 0.360) and retinal volume of the fovea (1-mm diameter around the fovea; *P* = 0.029, R = 0.332), parafovea (1–3-mm diameter; *P* = 0.002, R = 0.458), and macula (6-mm diameter; *P* = 0.003, R = 0.447). In the macular area (diameter: 6 mm), MPOD was correlated with the retinal neural volume of the ganglion cell layer (*P* = 0.037, R = 0.320), inner plexiform layer (*P* = 0.029, R = 0.333), and outer nuclear layer (*P* = 0.020, R = 0.353). Thus, MPOD may help in estimating neural health. Further studies should determine the impact of MP levels on neuroprotection.

## 1. Introduction

Macular pigment (MP), which is composed of lutein (L)/zeaxanthin (Z)/mezo-zeaxanthin (MZ) [1,2], is concentrated in the central part of the retina, the macula. MP absorbs short-wavelength light to prevent exposure of the retina to excessive light energy [3], and acts as an antioxidant in the retina [4]. In addition to animal experiments showing that lutein suppresses inflammatory mediators in the retina and reduces reactive oxygen species (ROS) by scavenging them and inducing antioxidative enzymes [5,6,7,8,9,10], epidemiological analyses conducted in the Age-related Eye Disease Study (AREDS) showed that dietary L/Z intake is inversely associated with the incidence of neovascular age-related macular degeneration (AMD), which can cause vision loss (odds ratio, 0.65); the incidence was lower in the high-carotenoid-intake (3.5 mg/day) group than in the low-carotenoid-intake (0.7 mg/day) group [11]. Moreover, the AREDS2 revealed that micronutrient supplementation with L/Z together with multi-vitamins and zinc can suppress the AMD progression rate. 

As humans cannot synthesize lutein because of a lack of the required enzymes, it must be ingested orally and absorbed by the intestine for delivery to each tissue/organ [12]. In the retina, MP is distributed in the inner and outer plexiform layers, where neural networks spread [13,14]. MP levels are evaluated in terms of the MP optical density (MPOD), which is reported to be positively related to dietary and serum lutein levels [5,14,15,16]. MPOD decreases with age, and MPOD of the fellow eye in AMD patients is lower than that of the fellow eye in individuals with no diseases other than cataracts [17,18]. Given that the number of retinal ganglion and photoreceptor cells decreases with age, [19,20] and even young healthy eyes have variations in visual function, as measured using techniques such as spatial-sweep steady-state pattern electroretinography, [21] eyes with no diagnosed retinal diseases may in fact have underlying retinal health-related conditions. Thus, the measurement of MPOD could provide additional information regarding such variations in retinal health among individuals. Donated brain tissue showed that MP carotenoid levels were significantly related to the brain L/Z levels in humans [22], suggesting that MPOD reflects brain L/Z levels. Considering that dietary L/Z intake improved cognitive performance in healthy adults in a double-blind, placebo-controlled trial [23], MPOD could be further applied for evaluating areas other than the retina, such as the brain. Nevertheless, while some previous reports showed a positive relationship between MPOD and central foveal thickness [24,25,26], others showed an inverse correlation between the juxtafoveal MPOD and retinal thickness [27]. Thus, the results are controversial, and the impact of MPOD on the condition of the retina remains unclear.

Optical coherence tomography (OCT) is an interferometric imaging modality that generates cross-sectional images by mapping the depth-wise reflections of low-coherence laser light from tissues [28]. OCT has improved the visualization of precise morphological features of the macula and is utilized for the diagnosis and assessment of the treatment responses of various macular diseases, including AMD, diabetic macular edema, and macular holes. Recent high-resolution OCT systems enable the calculation of the retinal volume in each retinal layer by generating three-dimensional OCT images; this is valuable for assessing the retinal neural condition in terms of characteristics such as synaptic and neural cell volumes [29]. Previous reports related to MPOD have focused on the overall retinal thickness. In the present study, which involved healthy adult volunteers, we evaluated whether MPOD is related to the retinal neural volume, determined which retinal layers showed an association with MPOD, and discussed whether MP could protect retinal neurons from daily stress.

## 2. Materials and Methods 

This study was conducted according to the guidelines of the Declaration of Helsinki. All procedures involving human subjects were approved by the Ethics Committee of Keio University School of Medicine (Approval No. 20150011), and the study was registered under the ID UMIN000017845. Informed consent was obtained from all subjects.

### 2.1. Subjects

This study was performed in the Medical Retina Division, Department of Ophthalmology, Keio University School of Medicine from January to December in 2017. Healthy Japanese volunteers without any ocular disease were considered eligible. Eyes with high myopia (<-6 diopters) were excluded. The final sample comprised 21 men and 22 women aged 22–48 years who agreed to provide clinical data.

### 2.2. MPOD Measurement

Absolute values of MPOD were measured using the macular pigment screener MPS2® (M.E. Technica Co., LTD, Tokyo, Japan), a macular densitometer that employs a heterochromatic photometry (HFP) technique, described elsewhere [30]. Briefly, the difference in the responsive intensity of blue- (absorbed by the MP), and green- (not absorbed by MP), wavelength flicker light in the fovea (where MP is concentrated) was compared with that in the parafovea (where MP is not concentrated) for measurement of the level of pigment that filtered blue-wavelength light in the fovea. We measured MPOD and the retinal volume for both eyes, and data of the eye with the higher MPOD without high myopia (<-6 diopters) was selected for further analyses. The data of right eyes without high myopia, and that of left eyes of subjects who had high myopia in the right eye but not in the left eye, were also analyzed for confirmation of the results.

### 2.3. Volumetric Analyses of the Macula and Retinal Layers Using OCT

OCT was performed using a Heidelberg Spectralis OCT system (Heidelberg Engineering GmbH, Dossenheim, Germany). OCT images were used to evaluate the central retinal thickness (CRT) and central choroidal thickness (CCT). CRT was defined as the distance between the internal limiting membrane and the presumed retinal pigment epithelium (RPE) at the fovea. CCT was defined as the distance between the hyper-reflective line corresponding to Bruch’s membrane beneath the RPE and the inner surface of the sclera at the foveal center, and was manually measured using the caliper function of the OCT device. Retinal volumetric and layer thickness analyses were performed using the three-dimensional recordings of the OCT images.

### 2.4. Ophthalmologic Examinations

All included subjects underwent best-corrected visual acuity measurements using the refraction test, intraocular pressure measurement, and fundus examination to confirm the absence of eye diseases.

### 2.5. Statistical Analyses

All results are expressed as the mean ± standard error (SE). Commercial statistical software (SPSS; ver. 25, SPSS Inc., IBM Corp, Armonk, NY, USA) was used for the analyses. The associations between MPOD and the retinal thickness and volume were assessed using multiple linear regression models after adjustment for age. Pearson’s correlation analyses were also performed. Differences were considered statistically significant at *P* < 0.05.

## 3. Results

The study included 43 healthy participants (range, 22–48 years; average age 31.4 ± 1.1 years; 21 men and 22 women, Table 1). The mean refraction, CRT, and CCT were -2.5 ± 0.3 diopter, 226 ± 2 μm, and 293 ± 15 μm, respectively. The mean MPOD was 0.589 ± 0.024 (Table 1) in the overall cohort, 0.625 ± 0.036 in men, and 0.555 ± 0.031 in women, with no significant differences between men and women (*P* = 0.149; data not shown). Confidence of MPOD levels was confirmed by the correlation of MPOD between the right and left eyes of the individuals (R = 0.806, *P* < 0.0001, 95% confidence interval [CI] 0.720 to 1.153, Figure 1A).

The choroid is the vessel-rich tissue that lies between the retina and sclera, and it supplies nutrients to the outer layers of the retina. Therefore, we analyzed the correlations between MPOD and the retinal and choroidal thicknesses. MPOD correlated positively with CRT (R = 0.360, *P* = 0.017, 95% CI 0.005 to 0.554, Figure 1B), but not with CCT (R = 0.014, *P* = 0.930, 95% CI -0.288 to 0.313, Figure 1C).

Next, we calculated the retinal neural volumes of the whole retinal layer. The average retinal volumes of the fovea (1-mm diameter around the fovea), parafovea (1–3-mm diameter), and macula (0–6-mm diameter) were 0.216 ± 0.002, 2.13 ± 0.01, and 8.66 ± 0.06 mm^3^, respectively (Table 2).

On analysis, we found that MPOD was positively correlated with the retinal neural volume of the fovea (*P* = 0.029, R = 0.332, 95% CI 0.036 to 0.575, Figure 2A), parafovea (*P* = 0.002, R = 0.458, 95% CI 0.183 to 0.666, Figure 2B), and macula (*P* = 0.003, R = 0.447, 95% CI 0.169 to 0.659, Figure 2C).

OCT can visualize the retinal layers in detail (Figure 3A), and it was utilized to calculate the retinal neural volume of each layer [31]. The average retinal neural volumes of the nerve fiber layer (NFL), ganglion cell layer (GCL), inner plexiform layer (IPL), inner nuclear layer (INL), outer plexiform layer (OPL), and outer nuclear layer (ONL) in the macular area were 1.01 ± 0.02, 1.06 ± 0.01, 0.87 ± 0.01, 0.96 ± 0.01, 0.84 ± 0.02, and 1.65 ± 0.03, respectively (Table 2). 

We analyzed the correlations between MPOD and the volume of each layer in the macular area. MPOD correlated with the retinal neural volumes of the GCL (*P* = 0.037, R = 0.320, 95% CI 0.022 to 0.566, Figure 3B); IPL, which is composed of neural networks (*P* = 0.029, R = 0.333, 95% CI 0.036 to 0.576, Figure 3C); and ONL, i.e., the photoreceptor layer (R = 0.353, *P* = 0.020, 95% CI 0.059 to 0.591, Figure 3D).

There were no correlations between age and ocular parameters, including CRT, CCT, and retinal volumes, and no differences between men and women with regard to these parameters in the current study (data not shown). Moreover, significant correlations were confirmed in the dataset of right eyes which did not have high myopia, together with those of the left eye in subjects who had high myopia in the right eye but not in the left eye (Appendix A).

Finally, we analyzed the correlations using multiple linear regression models (Table 3). After adjusting for age, which is reported to correlate negatively with MPOD [16,32,33,34], MPOD was positively correlated with CRT (*P* = 0.023, R = 0.351, 95% CI, 0.001 to 0.006) and the retinal neural volume of the fovea (*P* = 0.034, R = 0.326, 95% CI 0.287 to 7.077), parafovea (*P* = 0.002, R = 0.460, 95% CI 0.298 to 1.237,), macula (*P* = 0.003, R = 0.441, 95% CI 0.064 to 0.294), and the GCL (*P* = 0.049, R = 0.302, 95% CI 0.001 to 1.197,), IPL (*P* = 0.034, R = 0.326, 95% CI 0.067 to 1.619), and ONL (*P* = 0.017, R = 0.364, 95% CI 0.061 to 0.588, *P* = 0.017) of the macular area. 

## 4. Discussion

We demonstrated that MPOD was positively correlated with CRT and the retinal neural volumes of the fovea, parafovea, and macula. Moreover, MPOD was positively correlated with the retinal volumes of the GCL, IPL, and ONL in the macular area.

The mean MPOD in the current study was 0.59. This value was comparable to those measured using the HFP technique for young to middle-aged healthy individuals in our previous studies (0.67 [17] and 0.65 [16]) and other studies using other techniques (0.47 to 0.72 [35,36,37]).

We found a significant relationship between MPOD and CRT, consistent with previous studies on Chinese school-going children with low-to-moderate myopia [25] and Caucasian young and healthy adults [26]. More importantly, MPOD was positively correlated with the retinal volume, which directly represents the neural volumes of the retina, particularly those of the GCL, IPL, and ONL (the photoreceptor layer). Histological analysis has shown that lutein is distributed in the IPL, which is composed of neural networks involving synapses between the ganglion cells and inner layer neurons, and OPL, where photoreceptors from the ONL form synapses with the inner neurons [13,38]. Therefore, retinal layers that correlated with MPOD were consistent with lutein distribution in the retina. The absence of a correlation between MPOD and the OPL volume should be further analyzed in future studies. One possible explanation could be the precision of the OPL volume measurement, considering that OPL in the foveal region is very thin and could result in errors in adjustment of the measurement line in the OCT software. 

The MP lutein is shown to have neuroprotective effects in animal models. Sasaki et al. showed that regular dietary lutein intake attenuated synaptic loss and IPL thinning, as well as ganglion cell loss in diabetic mice [8], indicating that lutein intake protects neural components in the GCL and IPL. Moreover, lutein treatment prevented visual pigment (rhodopsin) reduction, outer segment shortening in photoreceptors, and photoreceptor dysfunction during retinal inflammation [7], while a lutein-supplemented diet attenuated visual function impairment after excessive light exposure by protecting against DNA damage in photoreceptor cells and preventing photoreceptor cell apoptosis [9]. Thus, lutein also protects photoreceptor cells. Greater MPOD and MP preservation may protect retinal neurons against everyday stress and improve the retinal neural condition. Therefore, the neural volume was greater when the tissue contained more MP and lutein. A similar effect could be hypothesized for the brain, considering that a clinical trial that has demonstrated a positive effect of L/Z intake on recognition ability [23]. However, further studies on this topic are required. 

Alternatively, a small retinal neural tissue volume may not pool large amounts of MP. Retinal diseases with a thin atrophic retina, including choroideremia [39], retinitis pigmentosa [40], oculocutaneous albinism [41], and glaucomatous eyes with foveal ganglion cell complex loss [42,43] showed low MPOD. However, the participants in the current study did not exhibit any of these retinal degenerative diseases.

This study was limited by the relatively small number of participants and the lack of power analyses. Moreover, all participants were young or middle aged; no older individuals or patients with AMD could be considered. Thus, all participants were healthy and had no retinal diseases or age-related effects. Nonetheless, our results indicated that MPOD and retinal neural volumes vary among young, healthy people. This is consistent with the finding in our previous study that young and healthy volunteers also exhibit variations in visual function, as detected using spatial-sweep steady-state pattern electroretinography [21]. Moreover, we only measured MPOD at 1-degree and did not evaluate the spatial distribution of MP; nonetheless, we found that MPOD was correlated with the parafoveal and macular volumes in addition to the foveal volume. This suggests that the measurement of MPOD at 1-degree could be sufficient to estimate the condition of the retina, including that of the parafovea and macula, although further analyses are required.

## 5. Conclusions

In summary, MPOD in healthy adults was correlated with the retinal neural thickness and volume. MP could play a role in preserving the neural tissue volume. Further studies are warranted for elucidating the impact of lutein in the neural tissue.

## Figures and Tables

**Figure 1 nutrients-12-00888-f001:**
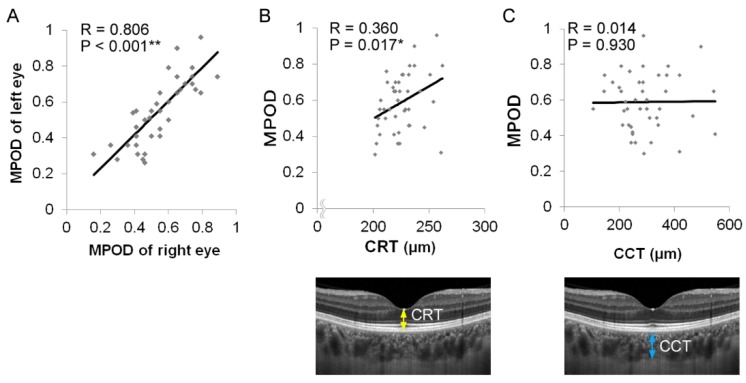
Correlation between the central retinal thickness (CRT), central choroidal thickness (CCT), and macular pigment optical density (MPOD). (A) Correlation of MPOD between the right and left eyes of the individuals was confirmed. (B) A positive correlation was seen between CRT and MPOD. The arrow in the OCT image indicates CRT. (C) No significant correlation was seen between CCT and MPOD. The arrow in the OCT image indicates CCT. ** *P* <0.01, * *P* <0.05.

**Figure 2 nutrients-12-00888-f002:**
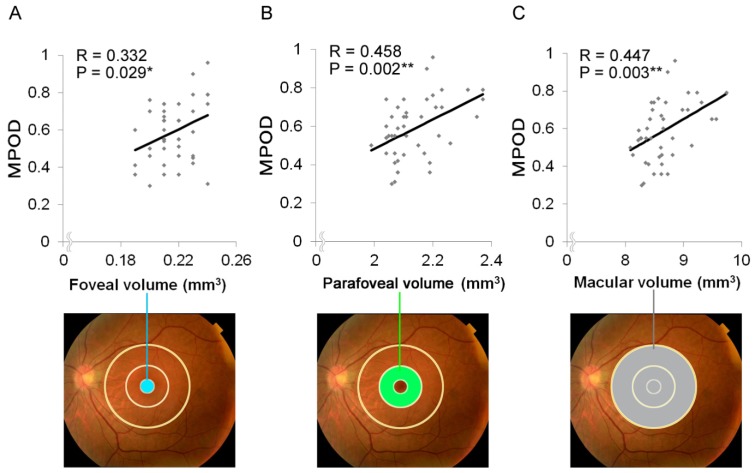
Correlations of the foveal, parafoveal, and macular volumes with the macular pigment optical density (MPOD). MPOD positively correlated with the retinal neural volumes of the fovea (1-mm diameter shown in blue; *P* = 0.029, R = 0.332, A), parafovea (1–3-mm diameter shown in green; *P* = 0.002, R = 0.458, B), and macula (6-mm diameter shown in gray; *P* = 0.003, R = 0.447, C). * *P* < 0.05, ** *P* < 0.01.

**Figure 3 nutrients-12-00888-f003:**
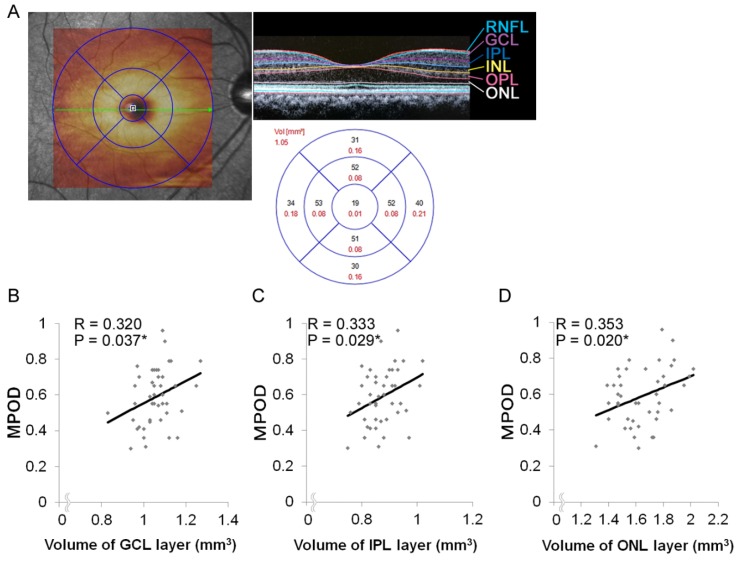
Correlation between the macular pigment optical density (MPOD) and volumes of retinal neural layers. (A) The macular area (left, the largest circle) was analyzed to calculate the volume of each retinal layer (right, upper), and the values obtained in each area (right, lower) were summed up to determine the values in the macular area (6-mm diameter). (B-D) MPOD was positively correlated with the retinal volumes of GCL (B), IPL (C), and ONL (D) in the macular area. RNFL retinal nerve fiber layer; GCL, ganglion cell layer; IPL, inner plexiform layer; INL, inner nuclear layer; OPL, outer plexiform layer; ONL, Outer nuclear layer; RPE, retinal pigment epithelium. * *P* < 0.05.

**Table 1 nutrients-12-00888-t001:** Participants characteristics.

*n* (eyes)	43
Age (years, mean, (range))	31.4 ± 1.1 (22–48)
Sex (male; eyes (%))	21 (48.8)
Refraction (diopter, mean, (range))	−2.5 ± 0.3 (−5.9–+1.4)
CRT (μm, mean, (range))	226 ± 2 (202–2620
CCT (μm, mean, (range))	293 ± 15 (104–548)
MPOD (log unit, mean, (range))	0.589 ± 0.024 (0.300–0.960)

Data are shown in mean ± SE. CRT, central retinal thickness; CCT, central choroidal thickness; MPOD, macular pigment optical density.

**Table 2 nutrients-12-00888-t002:** Retinal neural volumes measured from optical coherence tomography images.

	Average	Range
Fovea (0–1 mm, mm^3^)	0.216 ± 0.002	0.19–0.24
Parafovea (1–3 mm, mm^3^)	2.13 ± 0.01	1.99–2.37
Macula (0–6 mm, mm^3^)	8.66 ± 0.06	8.09–9.74
Nerve fiber layer (mm^3^)	1.01 ± 0.02	0.78–1.70
Ganglion cell layer (mm^3^)	1.06 ± 0.01	0.83–1.27
Inner plexiform layer (mm^3^)	0.87 ± 0.01	0.75–1.02
Inner nuclear layer (mm^3^)	0.96 ± 0.01	0.87–1.16
Outer plexiform layer (mm^3^)	0.84 ± 0.02	0.70–1.04
Outer nuclear layer (mm^3^)	1.65 ± 0.03	1.31–2.02

Data are shown as mean ± SE. The retinal volume of each layer was measured in the macular area.

**Table 3 nutrients-12-00888-t003:** Multiple linear regression analyses for the association between macular pigment optical density (MPOD) and retinal neural parameters.

	R	*P* Value	95% CI
CRT	0.351	0.023*	0.001 to 0.006
CCT	−0.012	0.942	−0.001 to 0.001
Retinal volume			
Retinal Areas			
Fovea (0–1 mm)	0.326	0.034*	0.287 to 7.077
Parafovea (1–3 mm)	0.460	0.002**	0.298 to 1.237
Macula (0–6 mm)	0.441	0.003**	0.064 to 0.294
Retinal Layers in the macular area			
Nerve fiber layer	0.111	0.482	−0.210 to 0.437
Ganglion cell layer	0.308	0.049*	0.001 to 1.197
Inner plexiform layer	0.326	0.034*	0.067 to 1.619
Inner nuclear layer	0.139	0.384	−0.425 to 1.080
Outer plexiform layer	0.142	0.365	−0.263 to 0.700
Outer nuclear layer	0.364	0.017*	0.061 to 0.588

Adjusted for age. CRT, central retinal thickness; CCT, central choroidal thickness; MPOD, macular pigment optical density. ** *P* <0.01, * *P* <0.05.

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
