# Peer review of "Correlation between Macular Pigment Optical Density and Neural Thickness and Volume of the Retina"

_nutrients, 2020, doi:10.3390/nu12040888_

Round 1
Reviewer 1 Report
The concerns of this reviewer have been addressed.
Author Response
The concerns of this reviewer have been addressed.
Thank you for your reviewing our manuscript and understanding our study.
Reviewer 2 Report
This is an interesting clinical research which focuses on, and provides current information on an age-long problem. The paper has improved significantly. I only have a few comments, below.
Abstract, line 20: The phrase “adult volunteers (21 men; age: 22–48 [average 31.4 ± 1.1] years)” appears incomplete because men’s was reported but not women. Please clarify. Similarly, results lines 118 - 119: “The study included 43 healthy participants (range, 22–48 years; average age 31.4 ± 1.1 years; 21 men)” is not clear. Please clarify.
Tables 1 and 3 footnotes, and Table 3 title should not be centered.
Author Response
This is an interesting clinical research which focuses on, and provides current information on an age-long problem. The paper has improved significantly. I only have a few comments, below.
Abstract, line 20: The phrase “adult volunteers (21 men; age: 22–48 [average 31.4 ± 1.1] years)” appears incomplete because men’s was reported but not women. Please clarify.
Thank you for your comment. The age data were for all the participants. We clarified number of women as follows;
Forty-three eyes of 43 healthy adult volunteers (21 men and 22 women; age: 22–48 [average 31.4 ± 1.1] years)...
Similarly, results lines 118 - 119: “The study included 43 healthy participants (range, 22–48 years; average age 31.4 ± 1.1 years; 21 men)” is not clear. Please clarify.
We revised as follows;
The study included 43 healthy participants (range, 22–48 years; average age 31.4 ± 1.1 years; 21 men and 22 women).
Tables 1 and 3 footnotes, and Table 3 title should not be centered.
We revised the points accordingly.
This manuscript is a resubmission of an earlier submission. The following is a list of the peer review reports and author responses from that submission.
Round 1
Reviewer 1 Report
This manuscript describes correlational and regression analyses between MPOD and the thickness and volume of retinal neural layers. On the positive side, this topic should be of some interest to the readers of Nutrients who study MPOD, lutein, zeaxanthin, and mesozeaxanthin, from a purely physiological standpoint - it's certainly helpful to know if MPOD is related to the structural features of the retina. However, while this research question is interesting, it is not novel and is a replication of previous work using (in this case) young to middle-aged healthy Japanese adults. I also have serious concerns about the statistical analyses presented, namely the use of logistic regression with a continuous criterion variable that does not seem to have been logit transformed. Finally, the introduction and discussion both need significant work in reviewing the previous literature on this topic and explaining the correlations seen in this study, as well as how they fit in with our broader understanding of the role of MP in retinal form and function.
I have gone through the manuscript in detail and noted the questions and concerns that I have in each section below. This manuscript needs extensive English language copy-editing to make it clear and comprehensible, so I have not included specific examples of sentence fragments, spelling/word choice errors, places where transition sentences are needed, etc. in my review below.
Abstract
- "MP may have a role in preserving neural tissue volume." While this may be true, the present correlational study is not capable of answering this question. Please either re-word to make it clear that this is an area for future research and not something that was analyzed in the present study or remove from the abstract.
- The abstract should make clear what specifically this study adds to the literature.
Introduction
- There are several potential reasons why macular pigment density would be related to retinal thickness or volume. For example, there are mechanical reasons related to the optics - OD is a function of the concentration of pigment but also its physical shape in the eye, which is affected by how pronounced (deep/shallow) the foveal depression is. Retinal volume is also measured here and volume would, again, be affected by the shape of the foveal pit where MP is heavily distributed. There are also prophylactic reasons why MPOD would be related to retinal thickness - an argument can be made that retinal thickness is an indicator of retinal health, so it could be the case that MP promotes a healthier retina, which in turn promotes a thicker retina. All of these are potential reasons why one would expect MPOD to be related to neural thickness and volume of the retina. These kinds of explanations should be clearly detailed, referencing the previous work that has been done on this, in the introduction. The authors reference two previous studies on this topic in the discussion (line 167) but it should be noted from the start, by referencing these (and other) works, that this is an interesting -- but not novel -- replication study using a different population. The introduction should make clear exactly what this study adds to the literature and why that gap in the literature needed to be filled.
- The relation between L/Z and MPOD and cognition is described in lines 47-51 but is never brought up again as far as I can see. This might be something that you could bring up again in the discussion as opportunities for future research and in relation to your significant correlation between GCL and MPOD. It seems like it should either be tied back in somewhere at the end or removed from the introduction.
- I would like to see a clearly stated hypothesis in the introduction, backed up by a thorough literature review.
Materials and Methods
- 2.1 - Subjects - I don't see any demographics information reported for your participants besides age and "gender". What about race? Also, I am assuming that when you report your participant's "gender" you actually mean their "sex", as "gender" is much more difficult to define and operationalize.
- 2.2 - Was MPOD measured only at 1-deg? or was the whole spatial distribution of MPOD measured?
- The HFP technique "described elsewhere" (citation 22) does not actually appear to be described in that paper. Please review this reference.
- 2.2 - If both eyes were measured, why were both eyes not analyzed? Why was the higher eye chosen? The rationale for these decisions needs to be discussed as it is currently unclear.
- 2.5 - citation(s) needed for the decision to adjust for age (it was a good idea, just need citations to back it up). It would also help to report correlations with age in support of using it as a control variable. Also, I assume that these variables of interest did not correlate with sex, given that it was not controlled for. If so, it would be helpful to note that and report the non-significant correlation coefficient. Also, did you check for correlations with race?
See the following papers for more information: https://www.sciencedirect.com/science/article/pii/0042698995002901
https://iovs.arvojournals.org/article.aspx?articleid=2188481
https://iovs.arvojournals.org/article.aspx?articleid=2184095 - 2.5 - Was a power analysis done? Was this sample size sufficient to detect small to moderate correlations? The small sample size is listed as a limitation but I don't see any discussion of a power analysis.
Results
- Table 1 needs "years" added as the unit of measure for Age.
- Table 1 needs "log units" added as the unit of measure for MPOD.
- Line 135-136 - this explanation of how "OCT can visualize retinal layers in detail" would be helpful sooner - perhaps just before line 118 where calculations of retinal neural volumes is first discussed.
- Lines 136-139 - all of the data discussed here on the volumes of each layer would be easier to understand in a table, or written out with each number directly after each layer. It is too hard to keep track of which number goes with which layer the way it is presented right now.
- In line 151 you say "Finally, we analysed the correlations using multiple linear regression models (Table 3)." Then, in Table 3 you present multivariate logistic regression analyses. It is unclear to me how you can analyze this data using logistic regression given that all of the variables are continuous, and therefore ORs don't seem to make sense. How can you have an OR for something that is not dichotomous? If you logit-transformed MPOD to fit a logistic regression analysis, this would need to be explained in detail, and I'm not sure would be appropriate. These analyses seem like they need to be re-done using multiple linear regression as was stated in the text.
- I am wondering if you have data on the full MP spatial distribution, as well as OCT images that you could use to calculate the morphology of the fovea, because comparing the morphology of the fovea with the MP spatial distribution, and including that in your models, would make this a much stronger and more novel study. If you only measured MPOD at 1-degree then this won't work, but I thought I'd include this in case you do have that data.
Discussion
- In lines 170-174 you provide a nice explanation for why MPOD would correlate with retinal volume in IPL and OPL, and state "Therefore, retinal layers which correlated with MPOD were consistent with lutein distribution in the retina." However, your results show that MPOD was only correlated with retinal thickness in the IPL, not the OPL. I think there is a lot of work that needs to be done in this section explaining why MPOD only correlated with certain layers, as well as why it did not correlate with CRT. What does that mean with regard to potential prophylactic explanations? One would think that MPOD would relate to choroid thickness if it were relating to a "healthier", thicker retina. Again, in lines 184 you state that "the neural volume was greater when the tissue involved more MP and lutein," but only 1 out of the 2 layers you discuss as being where MP is distributed correlated with MPOD.
Author Response
Point-by-Point Responses to the Reviewers’ Comments
We thank both reviewers for reviewing our manuscript and giving us helpful comments to improve our manuscript. We have addressed all the comment as described below.
Reviewer 1
I have gone through the manuscript in detail and noted the questions and concerns that I have in each section below. This manuscript needs extensive English language copy-editing to make it clear and comprehensible, so I have not included specific examples of sentence fragments, spelling/word choice errors, places where transition sentences are needed, etc. in my review below.
Thank you for your careful reviewing the constructive advice. Responses to each comment are as described below. We have asked English editing company “Editage” to edit the revised manuscript.
Abstract
- "MP may have a role in preserving neural tissue volume." While this may be true, the present correlational study is not capable of answering this question. Please either re-word to make it clear that this is an area for future research and not something that was analyzed in the present study or remove from the abstract.
According to your advice, we removed the sentence from the abstract.
- The abstract should make clear what specifically this study adds to the literature.
Previous reports have not shown the relationship between MP levels and three-dimensional neural retinal structure, so we revised the sentence as follows;
To evaluate whether MP levels are related to retinal neural protection and resulting health, we analyzed the association between the MP optical density (MPOD) and the macular thickness and volumes.
Introduction
- There are several potential reasons why macular pigment density would be related to retinal thickness or volume. For example, there are mechanical reasons related to the optics - OD is a function of the concentration of pigment but also its physical shape in the eye, which is affected by how pronounced (deep/shallow) the foveal depression is. Retinal volume is also measured here and volume would, again, be affected by the shape of the foveal pit where MP is heavily distributed. There are also prophylactic reasons why MPOD would be related to retinal thickness - an argument can be made that retinal thickness is an indicator of retinal health, so it could be the case that MP promotes a healthier retina, which in turn promotes a thicker retina. All of these are potential reasons why one would expect MPOD to be related to neural thickness and volume of the retina. These kinds of explanations should be clearly detailed, referencing the previous work that has been done on this, in the introduction. The authors reference two previous studies on this topic in the discussion (line 167) but it should be noted from the start, by referencing these (and other) works, that this is an interesting -- but not novel -- replication study using a different population. The introduction should make clear exactly what this study adds to the literature and why that gap in the literature needed to be filled.
Thank you for your advice. We added explanation to describe that this study filled the gap of previous controversial data that MPOD was reported to be positively or negatively correlated with retinal thickness, and therefore, we analyzed retinal volume in the three-dimensional OCT images. We also added that the retinal volume can represent retinal condition with reference 26, as follows;
However, previous reports showed a positive relationship between MPOD and the central foveal thickness 21 22 23, while others showed an inverse correlation between the juxtafoveal MPOD and the retinal thickness 24. Thus, the results are controversial, and the impact of MPOD on the condition of the retina remains unclear.….
…. Recent high-resolution OCT systems enable calculation of the retinal volume in each retinal layer by generating three-dimensional OCT images; this is valuable for estimating the retinal neural condition 26. Previous reports related to MPOD have focused on the retinal thickness. In the present study, which involves healthy adult volunteers, we evaluated whether MPOD is related to the retinal neural volume, determined the retinal layer that showed an association, and discussed whether MP could protect the retinal neurons from daily stress.
As regards relationship between MPOD and the shape of foveal pit, a previous report showed that the MPOD distribution was unrelated to the slope of the foveal pit (Westrup et al. 2014 DOI: 10.1167/iovs.13-12903). Also, we excluded highly myopic eyes that could influence MPOD measurement.
- The relation between L/Z and MPOD and cognition is described in lines 47-51 but is never brought up again as far as I can see. This might be something that you could bring up again in the discussion as opportunities for future research and in relation to your significant correlation between GCL and MPOD. It seems like it should either be tied back in somewhere at the end or removed from the introduction.
Thank you for your comment. We have included this point in the original manuscript in line 189-190 (line 207-209 in the revised manuscript) as follows;
A similar effect could be hypothesized for the brain, considering the clinical trial that has shown a positive effect of L/Z intake on recognition ability 20. However, further studies on this topic are required.
- I would like to see a clearly stated hypothesis in the introduction, backed up by a thorough literature review.
Thank you for your advice. We added a thorough literature review as described above, and revised to insert the hypothesis as follows;
In the present study, which involves healthy adult volunteers, we evaluated whether MPOD is related to the retinal neural volume, determined the retinal layer that showed an association, and discussed whether MP could protect the retinal neurons from daily stress.
Materials and Methods
- 1 - Subjects - I don't see any demographics information reported for your participants besides age and "gender". What about race? Also, I am assuming that when you report your participant's "gender" you actually mean their "sex", as "gender" is much more difficult to define and operationalize.
All the participants were Japanese in race, and involved in the revised version as below. We revised the word gender into sex.
Healthy Japanese volunteers without any ocular disease were considered eligible.
- 2 - Was MPOD measured only at 1-deg? or was the whole spatial distribution of MPOD measured?
Yes, it was measured only at 1-deg.
- The HFP technique "described elsewhere" (citation 22) does not actually appear to be described in that paper. Please review this reference.
We apologized for our mistake. Reference 22 is a paper on retinal structure and not for the HFP technique. We omitted the citation 22 in this part. Instead, we inserted citation 27 as follows;
Absolute values of MPOD were measured using the macular pigment screener MPS2® (M.E. Technica Co., LTD, Tokyo, Japan), a macular densitometer that employs a heterochromatic photometry (HFP) technique described elsewhere 27.
- 2 - If both eyes were measured, why were both eyes not analyzed? Why was the higher eye chosen? The rationale for these decisions needs to be discussed as it is currently unclear.
Thank you for your advice. Given that this is a subjective method, one cannot achieve greater values of MPOD than one’s ability, although one may show lower values than the ability if the participant did not respond well during the examination. Thus, we chose higher eye and compared with the OCT data of the same eye. We added this explanation in the revised version as follows;
This was because individuals cannot achieve MPOD values beyond their ability, although they may show lower values if they do not respond well during the examination, which is based on a subjective method.
- 5 - citation(s) needed for the decision to adjust for age (it was a good idea, just need citations to back it up). It would also help to report correlations with age in support of using it as a control variable. Also, I assume that these variables of interest did not correlate with sex, given that it was not controlled for. If so, it would be helpful to note that and report the non-significant correlation coefficient. Also, did you check for correlations with race?
See the following papers for more information: https://www.sciencedirect.com/science/article/pii/0042698995002901
https://iovs.arvojournals.org/article.aspx?articleid=2188481
https://iovs.arvojournals.org/article.aspx?articleid=2184095
Thank you for your advice. We adjusted for age because there were previous reports showing the negative correlation between MPOD and age, although we did not have significant correlation among the current participants (p = 0.440). We inserted references that show the negative correlation between MPOD and age, and revised the manuscript as follows;
Finally, we analyzed the correlations using multiple linear regression models (Table 3). After adjusting for age, which is reported to correlate negatively with MPOD 15 29 30 31,
There were no correlations between age and ocular parameters including CRT, CCT, and retinal volumes, and no sex differences in these parameters. we revised as follows;
The mean refraction, CRT, and CCT were -2.5 ± 0.3 diopter, 226 ± 2 μm, and 293 ± 15 μm, respectively. The mean MPOD was 0.589 ± 0.024 (Table 1) in the overall cohort, 0.625 ± 0.036 in men, and 0.555 ± 0.031 in women, with no significant differences between men and women (P=0.149; data not shown).
There were no correlations between age and ocular parameters, including CRT, CCT, and retinal volumes in the current study. Moreover, there were no sex differences in these parameters.
The participants were all Japanese as described above, and it was not necessary to adjust the race. We added description in the method part of the revised version as follows;
Healthy Japanese volunteers without any ocular disease were considered eligible.
- 5 - Was a power analysis done? Was this sample size sufficient to detect small to moderate correlations? The small sample size is listed as a limitation but I don't see any discussion of a power analysis.
No. We added this point in the limitation paragraph of the discussion as follows;
This study was limited by the relatively small number of participants and the lack of power analyses.
Results
- Table 1 needs "years" added as the unit of measure for Age.
We revised accordingly.
- Table 1 needs "log units" added as the unit of measure for MPOD.
We revised accordingly.
- Line 135-136 - this explanation of how "OCT can visualize retinal layers in detail" would be helpful sooner - perhaps just before line 118 where calculations of retinal neural volumes is first discussed.
Description of original line 135-136 was the explanation for the volume measurement of each retinal layer, and that of 118 was for the volume measurement of whole layers. For the readers to easily understand, we revised the original line 118 as follows;
Next, we calculated the retinal neural volumes of the whole retinal layers.
- Lines 136-139 - all of the data discussed here on the volumes of each layer would be easier to understand in a table, or written out with each number directly after each layer. It is too hard to keep track of which number goes with which layer the way it is presented right now.
The volumes of each layer were already shown in Table 2 in the original version, and in addition, we described the data also in the text. We moved the word (Table 2) from the previous sentence to this part for the readers to easily understand as follows;
The average retinal neural volumes of the nerve fiber layer (NFL), ganglion cell layer (GCL), inner plexiform layer (IPL), inner nuclear layer (INL), outer plexiform layer (OPL), and outer nuclear layer (ONL) in the macular area were 1.01 ± 0.02, 1.06 ± 0.01, 0.87 ± 0.01, 0.96 ± 0.01, 0.84 ± 0.02, and 1.65 ± 0.03, respectively (Table 2).
- In line 151 you say "Finally, we analysed the correlations using multiple linear regression models (Table 3)." Then, in Table 3 you present multivariate logistic regression analyses. It is unclear to me how you can analyze this data using logistic regression given that all of the variables are continuous, and therefore ORs don't seem to make sense. How can you have an OR for something that is not dichotomous? If you logit-transformed MPOD to fit a logistic regression analysis, this would need to be explained in detail, and I'm not sure would be appropriate. These analyses seem like they need to be re-done using multiple linear regression as was stated in the text.
Thank you for your pointing this out. We apologize for our mistake in the description. We have analyzed using multiple linear regression in the original manuscript, and we corrected this point in the revised manuscript.
Table 3. Multiple linear regression analyses for the association between macular pigment optical density (MPOD) and retinal neural parameters
- I am wondering if you have data on the full MP spatial distribution, as well as OCT images that you could use to calculate the morphology of the fovea, because comparing the morphology of the fovea with the MP spatial distribution, and including that in your models, would make this a much stronger and more novel study. If you only measured MPOD at 1-degree then this won't work, but I thought I'd include this in case you do have that data.
Thank you for your comment. We only measured MPOD at 1-degree, nonetheless, it was correlated with parafoveal and macular volume not only foveal volume. This might show that measurement of MPOD at 1-degree could be sufficient to estimate retinal volumes practically, although further analyses are required. We included this point in the discussion part as follows;
We only measured MPOD at 1-degree and not the spatial distribution of MP; nonetheless, we found that MPOD was correlated with the parafoveal and macular volumes in addition to the foveal volume. This suggests that the measurement of MPOD at 1-degree could be sufficient to estimate the condition of the retina, including the parafovea and macula, although further analyses are required.
Discussion
- In lines 170-174 you provide a nice explanation for why MPOD would correlate with retinal volume in IPL and OPL, and state "Therefore, retinal layers which correlated with MPOD were consistent with lutein distribution in the retina." However, your results show that MPOD was only correlated with retinal thickness in the IPL, not the OPL. I think there is a lot of work that needs to be done in this section explaining why MPOD only correlated with certain layers, as well as why it did not correlate with CRT. What does that mean with regard to potential prophylactic explanations? One would think that MPOD would relate to choroid thickness if it were relating to a "healthier", thicker retina. Again, in lines 184 you state that "the neural volume was greater when the tissue involved more MP and lutein," but only 1 out of the 2 layers you discuss as being where MP is distributed correlated with MPOD.
We agree that further studies are required to know why there was no correlation between MPOD and OPL volume in the current study. We added a potential prophylactic explanation in the discussion as follows;
The absence of a correlation between MPOD and the OPL volume should be further analyzed in future studies. One possible explanation could be the precision of the OPL volume measurement, considering that OPL in the foveal region is very thin and could result in errors in adjustment of the measurement line in the OCT software.
As regards CRT and choroidal thickness, we showed that there was a positive correlation between MPOD and CRT, and no correlation between MPOD and CCT in the original Table 3.

Reviewer 2 Report
In this manuscript, the authors describe a control/background study that asked if there was a relationship between density of macular pigment and retinal thickness/volume. They assessed this question in healthy individuals, measuring the optical density of the pigment as well as retinal layer thickness taken from OCT images. The authors find that the amount of pigment was positively correlated with central retinal thickness, foveal and macular volume, and volume of specific retinal layers (GCL, IPL, ONL). Overall, the methods are straightforward and the paper is well-written. Specific comments/concerns are given below.
- The authors had 43 participants and measured 43 eyes. Which eye was measured? Was it always the right or left eye?
- Please elaborate on ‘retinal neural volume’. Is this related to cell bodies or is it a thickness measurement? Figure 3A suggests it is maybe a length/thickness measurement. If it isn’t length, how was volume standardized, since some retinal layers have cell bodies and others do not (plexiform layers). ‘Volume’ of nuclear and plexiform layers may be comparable, but 'thickness' could be very different.
- The authors performed both correlation and regression analyses and the results were the same. Why do both analyses?
- The authors state (abstract) that the purpose of their study was ‘to determine the clinical significance of MP in the retina’. While this may be an ultimate goal, it wasn’t really the focus of the work and it isn’t what they did, since they only looked at healthy eyes. Rather, the paper seems more of a methods-type paper in which the authors show that MP can be easily measured clinically and that pigment levels (OD) reflect (predict?) retinal/foveal/macular volume. A comparison study using patients with ocular diseases would have been a nice addition and would get at the stated purpose. As it is now, the authors may want to revisit their purpose statement to indicate they have a clinically relevant non-invasive technique with high potential (or something similar).
Author Response
Point-by-Point Responses to the Reviewers’ Comments
We thank both reviewers for reviewing our manuscript and giving us helpful comments to improve our manuscript. We have addressed all the comment as described below.
Reviewer 2
Comments and Suggestions for Authors
In this manuscript, the authors describe a control/background study that asked if there was a relationship between density of macular pigment and retinal thickness/volume. They assessed this question in healthy individuals, measuring the optical density of the pigment as well as retinal layer thickness taken from OCT images. The authors find that the amount of pigment was positively correlated with central retinal thickness, foveal and macular volume, and volume of specific retinal layers (GCL, IPL, ONL). Overall, the methods are straightforward and the paper is well-written. Specific comments/concerns are given below.
Thank you for your careful reviewing and understanding our study.
- The authors had 43 participants and measured 43 eyes. Which eye was measured? Was it always the right or left eye?
Thank you for your comment.
As described in the original manuscript, MPOD of both eyes were measured and data of eyes with higher MPOD were utilized for further analyses including OCT analyses as follows;
We measured MPOD and the retinal volume for both eyes, and the eye with higher MPOD without high myopia (<-6 diopters) was selected for further analyses..
- Please elaborate on ‘retinal neural volume’. Is this related to cell bodies or is it a thickness measurement? Figure 3A suggests it is maybe a length/thickness measurement. If it isn’t length, how was volume standardized, since some retinal layers have cell bodies and others do not (plexiform layers). ‘Volume’ of nuclear and plexiform layers may be comparable, but 'thickness' could be very different.
We apologize for the confusing figure. The original Figure 3A was the image for showing the retinal layers. Retinal volumes were measured in three-dimensional OCT images using built-in software, and standardaization of the volume was not necessary. We revised the Figure 3A (see below) for the readers to easily understand.
Legend
- A) The macular area (left, the largest circle) was analyzed to calculate the volume of each retinal layer (right, upper), and the values obtained in each area (right, lower) were summed up to determine the values in the macular area (6-mm diameter).
- The authors performed both correlation and regression analyses and the results were the same. Why do both analyses?
We and others have previously reported that MPOD and age had a negative correlation. Thus, we adjusted for age. The results were the same, therefore the relationship between MPOD and retinal thicknesses and volumes were confirmed.
- The authors state (abstract) that the purpose of their study was ‘to determine the clinical significance of MP in the retina’. While this may be an ultimate goal, it wasn’t really the focus of the work and it isn’t what they did, since they only looked at healthy eyes. Rather, the paper seems more of a methods-type paper in which the authors show that MP can be easily measured clinically and that pigment levels (OD) reflect (predict?) retinal/foveal/macular volume. A comparison study using patients with ocular diseases would have been a nice addition and would get at the stated purpose. As it is now, the authors may want to revisit their purpose statement to indicate they have a clinically relevant non-invasive technique with high potential (or something similar).
Thank you for your comment. To analyze diseased eyes would an interesting future study. We revised the abstract to show the goal of the current study as follows;
To evaluate whether MP levels are related to retinal neural protection and resulting health, we analyzed the association between the MP optical density (MPOD) and the macular thickness and volumes.
We also revised the introduction as follows;
Recent high-resolution OCT systems enable calculation of the retinal volume in each retinal layer by generating three-dimensional OCT images; this is valuable for estimating the retinal neural condition 26. Previous reports related to MPOD have focused on the retinal thickness. In the present study, which involves healthy adult volunteers, we evaluated whether MPOD is related to the retinal neural volume, determined the retinal layer that showed an association, and discussed whether MP could protect the retinal neurons from daily stress.

Round 2
Reviewer 1 Report
I thank the authors for their work in editing this paper but find myself with more concerns than I believe they will be able to address in a simple revision. For example:
-Confidence levels of MPOD measurements should be reported in some way (e.g., variability between eyes within subjects, scatterplot showing correlation between OD and OS within subjects).
-I am still unclear why the "higher eye" was chosen. The explanation that was provided in this revision - that individuals "cannot achieve MPOD values beyond their ability" - is factually inaccurate. It is entirely possible for subjects to perform in such a way on the task that MPOD levels appear to be higher than they actually are. Therefore, it would be helpful to see information reported via the MPS2 about the confidence of each subject's MPOD measurement. The fact that the authors believe that subjects cannot perform in such a way that their recorded MPOD is higher than it actually is concerns me.
-The authors need to specifically state which MPS2 mode was used for the MPOD measurements. If it was anything but the central and parafoveal mode, then the study is invalid as it has been well documented in the literature that a parafoveal reference is necessary for accurate measurements.
-I would like to see a brief statement regarding the method that was used to assess that participants had "healthy" eyes. It may be assumed by most in the field, but should still be laid out clearly (e.g., slit lamp, fundus opthalmoscopy, etc.).
-There are errors in wording throughout this manuscript that concern me. For example, (line 87) MP does not "block" blue light - blocking and filtering are very different things. Also, (line 88) photoreceptors don't have a "threshold for recognition" - the brain certainly does, but photoreceptors do not.
-The authors refer to the "retinal neural condition" throughout this paper (e.g., line 65 - "estimating the retinal neural condition") and I'm not exactly sure what that means.
-The authors say that they wrote "logistic" regression in Table 3 in error, but then reported odds ratios in the table and did not change that at all in the revision. I don't know how one would get odds ratios with the data presented (both continuous variables) without logit transforming one of them, and that was not discussed here.
I will give the authors the benefit of the doubt and assume that their work is not being represented well in this paper due to extensive English language issues, but I then have to suggest that they take the time needed to get help with the editing that is needed from an expert in the field who also is an English language speaker, as I think the technical terms are not translating appropriately. There are statements throughout the paper that, while being grammatically correct, don't make any sense from a scientific standpoint. Another example that gives me concern is their response that "Description of original line 135-136 was the explanation for the volume measurement of each retinal layer, and that of 118 was for the volume measurement of whole layers." The difference between "each retinal layer" and "whole layers" eludes me.
Again, the science behind this paper may be sound, but the presentation and explication leaves me with enough concern that it may not be.
Author Response
Comments and Suggestions for Authors
I thank the authors for their work in editing this paper but find myself with more concerns than I believe they will be able to address in a simple revision. For example:
-Confidence levels of MPOD measurements should be reported in some way (e.g., variability between eyes within subjects, scatterplot showing correlation between OD and OS within subjects).
Response: Thank you for your comment. Accordingly, we have added a scatter plot showing the correlation between right and left eye data of individuals (Figure 1A). The revised text and figure are as follows:
Confidence of MPOD levels was confirmed by the correlation of MPOD between right and left eyes of individuals (R = 0.806, P < 0.0001, 95% confidence interval [CI] 0.720 to 1.153, Figure 1A).
Revised Figure 1 legend
(A) Correlation of MPOD between right and left eyes of individuals was confirmed.
-I am still unclear why the "higher eye" was chosen. The explanation that was provided in this revision - that individuals "cannot achieve MPOD values beyond their ability" - is factually inaccurate. It is entirely possible for subjects to perform in such a way on the task that MPOD levels appear to be higher than they actually are. Therefore, it would be helpful to see information reported via the MPS2 about the confidence of each subject's MPOD measurement. The fact that the authors believe that subjects cannot perform in such a way that their recorded MPOD is higher than it actually is concerns me.
Response: As described above, information about the confidence of each subject’s MPOD measured using MPS2 has been provided by adding the revised Figure 1A. The MPOD values were highly correlated between right and left eyes; as a rule, in this situation we decided to choose the value of the eye which showed a higher MPOD value. We had earlier considered that if the participants did not concentrate on the examination, their MPOD could have been underestimated, but we have deleted that text from the manuscript. Instead, we showed the correlation of the MPOD between right and left eyes in Figure 1A as described above.
In addition, the results were confirmed by analysis of data of right eyes and left eyes only when the right eyes were highly myopic, and shown in Supplementary Figure 1. The corresponding sections of text were revised as follows:
Methods
The data of right eyes without high myopia and that of left eyes of subjects who had high myopia in the right eye but not in the left eye were also analyzed for confirmation of the results.
Results
Moreover, significant correlations were confirmed in the dataset of right eyes which did not have high myopia together with those of the left eye in subjects who had high myopia in the right eye but not in the left eye (Supplementary Figure 1).
Supplementary Figure 1
Correlations between MPOD and CRT, foveal, parafoveal, and macular volumes, and GCL, IPL, and ONL volumes of the macular area from the dataset of right eyes which did not have high myopia together with those of the left eye in subjects who had high myopia in the right eye but not in the left eye. MPOD, macular pigment optical density; GCL, ganglion cell layer; IPL, inner plexiform layer; ONL, outer nuclear layer. *P < 0.05.
-The authors need to specifically state which MPS2 mode was used for the MPOD measurements. If it was anything but the central and parafoveal mode, then the study is invalid as it has been well documented in the literature that a parafoveal reference is necessary for accurate measurements.
Response: We used absolute MPS2 value mode, which were evaluated by measuring central and parafoveal values and utilizing the parafoveal value as reference. We had included this point in the original manuscript; furthermore, in accordance with your advice below, we have also changed the words “blocked” to “filtered” and “recognition” to “responsive” as follows:
Absolute values of MPOD were measured using the macular pigment screener MPS2® (M.E. Technica Co., LTD, Tokyo, Japan), a macular densitometer that employs a heterochromatic photometry (HFP) technique described elsewhere 27. Briefly, the difference in the responsive intensity of blue (absorbed by the MP)- and green (not absorbed by MP)-wavelength flicker light in the fovea (where MP is concentrated) was compared with that in the parafovea (where MP is not concentrated) for measurement of the level of pigment that filtered blue-wavelength light in the fovea.
-I would like to see a brief statement regarding the method that was used to assess that participants had "healthy" eyes. It may be assumed by most in the field, but should still be laid out clearly (e.g., slit lamp, fundus opthalmoscopy, etc.).
Response: We have included Ophthalmic Examinations in the original manuscript and have added the last phrase (highlighted in yellow) in the latest version as follows:
All included subjects underwent best-corrected visual acuity measurements using the refraction test, intraocular pressure measurement, and fundus examination to confirm the absence of eye diseases.
-There are errors in wording throughout this manuscript that concern me. For example, (line 87) MP does not "block" blue light - blocking and filtering are very different things. Also, (line 88) photoreceptors don't have a "threshold for recognition" - the brain certainly does, but photoreceptors do not.
Response: According to your advice, we have revised the corresponding text as follows:
Briefly, the difference in the responsive intensity of blue (absorbed by the MP)- and green (not absorbed by MP)-wavelength flicker light in the fovea (where MP is concentrated) was compared with that in the parafovea (where MP is not concentrated) for measurement of the level of pigment that filtered blue-wavelength light in the fovea.
-The authors refer to the "retinal neural condition" throughout this paper (e.g., line 65 - "estimating the retinal neural condition") and I'm not exactly sure what that means.
Response: Thank you for pointing this out. We realize that at certain instances, the term could have been better explained or phrased; accordingly, we have added some explanations and reworded the phrase, depending on the context, as follows:
Given that the number of retinal ganglion and photoreceptor cells decreases with age, 19 20 and even young healthy eyes have variations in visual function as measured using techniques such as spatial-sweep steady-state pattern electroretinography, 21 eyes with no diagnosed retinal diseases may in fact have underlying retinal health-related conditions. Thus, the measurement of MPOD could provide additional information regarding such variations in retinal health among individuals.
For line 65 in the previous version, we have revised as follows:
Recent high-resolution OCT systems enable calculation of the retinal volume in each retinal layer by generating three-dimensional OCT images; this is valuable for assessing the retinal neural condition in terms of characteristics such as synaptic and neural cell volumes 26.
-The authors say that they wrote "logistic" regression in Table 3 in error, but then reported odds ratios in the table and did not change that at all in the revision. I don't know how one would get odds ratios with the data presented (both continuous variables) without logit transforming one of them, and that was not discussed here.
Response: We sincerely apologize for this error, as “OR” should also have been revised to “R”. We have now made the corresponding correction in Table 3.
I will give the authors the benefit of the doubt and assume that their work is not being represented well in this paper due to extensive English language issues, but I then have to suggest that they take the time needed to get help with the editing that is needed from an expert in the field who also is an English language speaker, as I think the technical terms are not translating appropriately. There are statements throughout the paper that, while being grammatically correct, don't make any sense from a scientific standpoint. Another example that gives me concern is their response that "Description of original line 135-136 was the explanation for the volume measurement of each retinal layer, and that of 118 was for the volume measurement of whole layers." The difference between "each retinal layer" and "whole layers" eludes me.
Again, the science behind this paper may be sound, but the presentation and explication leaves me with enough concern that it may not be.
Response: The phrase “each retinal layer” was meant to refer to the layers in the macular area, such as the nerve fiber layer, ganglion cell layer, and so on. The term “whole layer” is meant to refer to the entire retina between the internal limiting membrane and the presumed retinal pigment epithelium (RPE) and to include all the retinal layers.
According to your advice, we have had the manuscript edited by an English language editor with ophthalmological experience.